# Effect of Flour Particle Size on the Glycemic Index of Muffins Made from Whole Sorghum, Whole Corn, Brown Rice, Whole Wheat, or Refined Wheat Flours [note 1]

**DOI:** 10.3390/foods12234188

**Published:** 2023-11-21

**Authors:** Ashley Pruett, Fadi M. Aramouni, Scott R. Bean, Mark D. Haub

**Affiliations:** 1Department of Food, Nutrition, Dietetics and Health, Kansas State University, Manhattan, KS 66506, USA; apruett87@gmail.com (A.P.); haub@k-state.edu (M.D.H.); 2Center for Grain and Animal Health Research, USDA-ARS, Manhattan, KS 66502, USA; scott.bean@usda.gov

**Keywords:** sorghum, glycemic index, flour particle size, wheat, corn, whole grain, rice

## Abstract

The unique properties of sorghum are increasingly being studied for potential health benefits, with one area of emphasis being the impact of sorghum consumption on mitigating type 2 diabetes. The glycemic index (GI) of muffins made from whole grain sorghum flour ground to three different particle sizes (fine, intermediate, coarse) was tested on eight healthy volunteers (ages 18–40) and compared to the glycemic index of whole grain corn, wheat, and rice flours produced using a similar product formula. Sorghum flour ground through a 0.5 mm screen (“fine”) had an overall similar particle size to that of the brown rice flour ground using a 0.5 mm screen. The range of GI values was 32 to 56, with only the GI of intermediate milled sorghum flour being lower than that of corn, rice, or wheat (*p* < 0.05). The lowest glycemic index (32 +/− 17) was found when using sorghum flour with an intermediate particle size (167 +/− 4 μm). Muffins made using brown rice had the next lowest glycemic index at 37 +/− 17. All GI values calculated had large standard deviations, which is common for these types of studies. These results can assist in the product development process to advance the quality of healthy, gluten-free sorghum-based foods for consumers. Further research should investigate if these results can be duplicated and the possible reason for the lower GI of intermediate particle size sorghum flour.

## 1. Introduction

Cereal grains are important food staples around the world and a major source of calories and nutrients for a significant part of the world population [1]. In many countries, health problems exist that are, to various extents, related to lifestyle, one of which includes diet. Dietary deficiencies can lead to, or exacerbate, conditions such as obesity, diabetes, cardiovascular health, certain cancers, and contribute to overall morbidity [2,3]. 

Grains, especially whole grains, contain a wide range of beneficial compounds for human health [4,5,6]. Consumption of whole grains has been reported to be related to prevention of cardiovascular disease, high blood pressure, certain cancers, obesity, and diabetes [7,8,9,10,11,12]. Consumption of whole grains may provide human health benefits through several mechanisms including delivery of antioxidants, anti-inflammatory compounds, resistant starch, fiber, vitamins, and minerals, which are most often found in the bran of whole grains [4,13,14,15]. 

The impact of consumption of whole grains has been related to preventing and managing type 2 diabetes, which may at least be partially related to the rate of digestion of carbohydrates and subsequent effects on insulin and blood sugar levels [5,16,17]. The glycemic index (GI) is one index of how fast carbohydrates of a particular food are enzymatically broken down in the human body, which in turn affects the insulin levels produced in the pancreas and circulating in the blood. Several factors have been shown to impact the GI of foods, including fiber types and levels, starch structure, presence of phenolic compounds, food structure, and flour particle size [16,18,19]. Other properties of low-GI whole-grain foods that may be beneficial in mitigating type 2 diabetes include increased feelings of satiety, beneficial impacts on gut microbiome, and the presence of bioactive compounds [16,17].

Although sorghum has been grown for years in Africa and many parts of Asia, consumption of sorghum-based food products in the United-States has recently received increased attention, primarily due to its status as a gluten-free cereal [20,21] and due to its high levels of phenolic compounds [22]. This has led to an increase in the potential for innovative sorghum-based foods [23]. The anti-inflammatory activity of select sorghum brans has been studied and confirmed since 2010 [24] and more recently reported for sorghum aqueous and ethanolic extracts [25]. Moreover, there is a consensus among scientists that phenolic compounds in whole-grain sorghum offer a variety of health benefits [26], including tumor suppression in colon cancer models [27]. The antidiabetic effects of three Korean sorghum phenolic extracts have been studied in diabetic rats [28]. Several recent reviews have highlighted the components of sorghum grain and their potential human health benefits, including benefits in mitigating and helping to prevent type 2 diabetes through a number of possible mechanisms [29,30,31,32,33,34]. Simnadis et al. [30] conducted a systematic literature review of studies investigating the effect of the consumption of sorghum and its potential benefits on human subjects and reported that the consumption of sorghum may contribute to improvements in insulin and blood sugar levels.

While several factors can influence the GI of foods, flour particle size is one factor that may influence the digestibility and subsequent GI of grain-based foods [32,33,34,35]. Very little published research was found on the effect of sorghum flour particle size on the GI of sorghum in human subjects. Therefore, given the potential for consumption of whole-grain sorghum food products to help control blood sugar and insulin responses, the goal of this research was to test the effect of flour particle size on the GI of a whole-grain sorghum muffin and compare this GI to similar products made from other whole-grain flours.

## 2. Materials and Methods

### 2.1. Grain Samples and Flour Preparation

Whole sorghum grain (*Sorghum bicolor*; white-grained variety Fontanelle 1000), hard red winter wheat (*Triticum aestivum*; Karl 92), yellow corn (*Zea mays*; Dynagro 57V15), a commercial sample of brown rice (*Oryza sativa*; purchased from HyVee Inc., West Des Moines, IA, USA), and commercial all-purpose wheat flour (HyVee Inc., West Des Moines, IA, USA) were used for this study. Whole grains were ground into flour using a UDY Cyclone Lab Sample Mill (UDY Corp., Fort Collins, CO, USA). Sorghum grain was milled into three samples using a 0.5 mm screen (fine), a 1 mm screen (intermediate), and a 2 mm screen (coarse). Wheat, brown rice, and corn grains were ground using a 0.5 mm screen. All seven flour samples were placed in re-sealable bags (Ziploc Brand, New Brunswick, NJ, USA) and stored in a commercial freezer at approximately −3 °C until they were used.

### 2.2. Flour Characterization 

Flour particle size was measured using a Beckman Coulter LS™ 13 320 SW Dry Powder System Laser Diffraction Particle Size Analyzer (Beckman-Coulter, Inc., Miami, FL, USA) to determine the particle size distribution of the milled flours. The flour was placed into the load cell until it was approximately 2/3 full. The cell was then loaded into a Tornado™ Dry Powder Dispersing attachment for the instrument, and measurements were taken. 

Starch damage was determined using a Megazyme Starch Damage Assay Procedure, K-SDAM 02/2008 with a Limit of Detection of 0.5 g/100 g (Megazyme International Ireland Ltd., Co., Wicklow, Ireland), following AACC-approved methods 76-31.

### 2.3. Muffin Formulation and Preparation

All seven muffin formulations were composed of 300 g of flour, 4.5 g of baking powder (Clabber Girl, Terre Haute, IN, USA), 3.5 g of salt (Morton, Chicago, IL, USA), and tap water. Water levels varied on account of flour type and particle size and were based on test bakes to determine optimal water levels that resulted in muffins deemed unanimously acceptable in terms of texture by the 8 human subjects who were recruited to consume them. Amounts of water used per formula are listed in Table 1. 

Dry ingredients (flour, baking powder, and edible salt) were measured and mixed by hand for 30 s. Water was added to the dry mixture and blended by hand for 1 min. Sixty grams of batter was scaled out into each cup of a twelve-cup muffin tray. No oil was applied to muffin trays before baking. Muffins were baked in a pre-heated oven at 177 °C. Muffins made from all-purpose flour, whole wheat flour, corn flour, and rice flour were baked for 30–35 min, whereas those made from sorghum flour were baked for 15 min (baking times were determined in preliminary experiments to determine optimal times). Muffins were cooled in the tray for ~10 min, and then removed and placed on a cooling rack for an additional 2 h, or until muffins reached room temperature. Muffins were placed in re-sealable bags (Ziploc Brand, New Brunswick, NJ, USA), labeled, and stored in a commercial freezer at approximately −3 °C until human trials.

### 2.4. Proximate Analysis 

The moisture contents of the flours were measured using the Association of Official Analytical Chemists (AOAC)-approved method 930.15. The protein content of each flour was measured using AOAC-approved method 990.03, with a nitrogen to protein conversion factor of 6.25. The crude fat content of each flour was measured using AOAC-approved method 920.39. The crude fiber content of each flour was measured using the Ankom Method, based on AOAC 962.09, and ash contents were measured using AOAC-approved method 942.05 on an “as is” basis. Total carbohydrate for each muffin type was calculated by difference using results from the above proximate analysis. Muffins were then formulated to contain 20 g of available carbohydrate using the following calculation:$$Muffin\;treatement\;weight\;\left(g\right)=\frac{total\;carbohydrate\;(g)}{20\;g\;available\;carbohydrate}\times 100\;g\;sample$$

The final serving size for each muffin is shown in Table 2.

### 2.5. Evaluation of Glycemic Index

The seven treatments included muffins made from all-purpose flour, whole wheat flour, brown rice flour, corn flour, and sorghum flours with three different particle sizes (fine, intermediate, and coarse). Treatments were labeled with codes 101–107.

#### 2.5.1. In Vivo Protocol

Eight healthy volunteers (aged 18–40) participated in this study. Volunteers were recruited among Kansas State University students. All subjects gave their signed consent and were aware of all considerations surrounding the study. Approval for this study was obtained from the Institutional Review Board (IRB# 3941.2). Subjects were asked to maintain their regular lifestyle throughout the entirety of the study, avoiding any extreme behaviors. Each subject selected two weekly time slots for their testing. Subjects received two treatments per week in random order with at least a 48 h period in between. Each treatment was given twice as well as one dose of a 20 g dextrose drink (reference food) over an eight-week period. Subjects were required to fast for ten hours prior to each visit but were permitted to consume unlimited drinking water throughout testing. A weighed portion of muffin containing 20 g of available carbohydrate that had been thawed to room temperature overnight was consumed at each testing session. Two finger prick samples were taken from volunteers for capillary blood analysis by a lab assistant at 0 (fasting), 30, 45, 60, 90, and 120 min after consumption, and then averaged. A YSI 2300 Glucose Oxidase Analyzer (Yellow Spring’s Instruments, Yellow Springs, OH, USA) was used to measure blood glucose concentrations.

#### 2.5.2. Calculation of Glycemic Index

The incremental area under the glycemic response curve (iAUC) was constructed using the trapezoid model, with fasting levels as the baseline, using GraphPad Prism 5 software (GraphPad Software, Inc., La Jolla, CA, USA). Area below the baseline was excluded. The GI values were calculated by dividing the iAUC for the test food (muffin) by that for the standard (20 g dextrose drink) and multiplied by 100. The GI for each treatment was calculated as the mean from the respective average GIs of the eight volunteers. Glycemic index was calculated using the following formula: $$Glycemic\;index=\frac{iA\;C\;\left(treatment\right)}{iA\;C\;(standard)}\times 100$$

### 2.6. Statistical Analysis

Two replicates of each treatment were used in a randomized block design. GI values were analyzed using SAS, Software Release 9.3 (SAS, Institute Inc., Cary, NC, USA, 2011). The differences in response between grains, flour particle size, and the proximate analysis composition of muffins were analyzed for significance. When treatment effects were found to be significantly different, the least square means with Dunnett’s *p*-values were used to differentiate treatment means. A level of significance was reported at *p* < 0.05.

Regression analysis was performed to find out whether a correlation between starch damage and glycemic response exists, using Excel (Microsoft Office Excel, Microsoft Corporation, Redmond, WA, USA).

## 3. Results and Discussion

### 3.1. Flour Particle Size

The particle size of flour represents the degree of volume of particles and the total exposed surface area of particles dispersed throughout the flour [36]. For the sorghum flours ground using three different screen sizes, particle size varied as expected, with mean d90 value and mean particle size distribution increasing as screen size increased (Table 3). The d90 value represents the size where 90% of the volume of flour particles is less than the given values in microns, and the mean particle size distribution represents the overall average particle size of the flour. Sorghum flour ground through the 0.5 mm screen (“fine”) had an overall similar particle size to the brown rice flour ground using the 0.5 mm screen. Corn and wheat ground using the 0.5 mm screen had similar d90 values, while wheat had the largest mean particle size distribution among the samples ground using the 0.5 mm screen. The particle size distribution of a given flour is the result of kernel hardness, moisture content, kernel mass and milling methods [37]. These differences likely reflect differences in grain hardness and grain structure among the samples. 

### 3.2. Flour Properties, Proximate Composition, and Relationships to GI

To provide additional information on the GI responses among the flours tested in this study, proximate analysis was conducted, and the properties of the flours were characterized. The proximate composition for the flours is shown in Table 4. Although values for total carbohydrate in Table 2 and Table 3 differ, all muffin treatments were administered as 20 g of available carbohydrate. Whole wheat muffins contained the highest level of protein, and corn muffins had the highest percentage of fat content (Table 2 and Table 4). Both protein and fat can influence the GI of foods but were not major factors in this study, and corn and wheat had higher GI levels than those of rice and the intermediate particle size of sorghum flour. Muffins made from the coarsely milled sorghum flour had significantly higher fiber than all other muffins except for the whole wheat muffins (Table 2 and Table 4). This is an inconsistent and unexpected relationship, as all sorghum flours in this study were milled from the same lot of sorghum. The different particle sizes of the sorghum flours could have caused a difference in reaction rates or in the efficacy of the analytical methods used. Since the serving sizes of the test muffins were calculated from these results, differences between proximate analysis test results may have impacted the GI results to some degree.

When grains are ground or milled into flour, intact starch granules become damaged, leaving starch granules that have been fractured, shattered, or chopped [38]. Damaged starch readily absorbs more water than intact starch, but its effect on the GI of grains has not yet been clearly determined. Grain hardness is a major contributor to starch damage that results from milling. In soft wheat, starch granules are loosely bound within the kernel and are easily released, whereas starch granules in hard wheat are tightly bound within the protein matrix and are much more susceptible to damage if the endosperm is fractured [39]. One would also assume that the less abrasion applied to a grain during milling, the less final starch damage would result. In the current study, as the particle size of the sorghum flours increased, starch damage decreased (Figure 1). Brown rice flour exhibited the highest percentage of starch damage among all flours (Figure 1).

### 3.3. GI

Among the muffins made using whole-grain sorghum flour, the flour with the intermediate particle size had the lowest GI (Table 5). The range of values was 32 to 56, with only the GI of intermediate milled sorghum flour being smaller than that of corn, rice, and wheat (*p* < 0.05). Values for the glycemic index of sorghum and sorghum products reported in the literature vary greatly. This could be due to sorghum grain variety, milling method, processing conditions, or issues with GI testing itself. The GI of sorghum is reported to lie between 72 and 70 by the Diabetes Council [40]. Sydney University’s Glycemic Index Research Service [41] reported GI values for sorghum foods ranging from 54 to 85. Shobana et al. [42] tested the GI of 12 foods including sorghum-based foods and found that 5 were in the high GI category (finger millet balls, sorghum, pearl millet, and maize roti), 4 in the medium GI category (sorghum idli, wheat dosa, methi roti, and adai), and 3 in the low GI category (broken wheat upma, white peas sundal, and white chickpeas sundal). 

Moraes et al. [43] studied the estimated glycemic index (EGI) of sorghum bran (SB), decorticated sorghum flour (DSF), and whole sorghum flour (WSF). They reported EGI values of 84.5 ± 0.41, 77.2 ± 0.33, and 60.3 ± 0.78 for DSF, WSF, and SB, respectively. Phenolic compounds, antioxidant activities, total insoluble and soluble dietary fiber, and β-glucans of sorghum flour samples were all negatively correlated to EGI, but resistant starch content was not.

Wolter et al. [44] investigated the effect of sourdough processing on gluten-free breads and reported that sourdough treatment reduced GI for sorghum breads. Prasad et al. [45] tested the GIs of some sorghum-based foods and found them to be significantly lower than those of their respective wheat-/rice-based foods. It is obvious that reported data on the GI of sorghum vary greatly, indicating the need for more research in this area.

As particle size decreases, more surface area is exposed to digestive enzymes, and this may lead to a higher glycemic response as a result. Holt and Miller [46] showed a higher glycemic response to wheat as particle size decreased by testing equal carbohydrate portions of baked whole grains and cracked grains and muffins from coarse and fine whole meal flours. Two studies by O’Dea et al. [47], and Collier and O’Dea [48] compared glucose responses to whole brown rice and ground brown rice and concluded that the actual form of complex carbohydrate is critical in determining the metabolic response. Ground brown rice elicited significantly higher glucose responses in both cases in those studies. Particle size has not always been found to influence GI; for example, Behall et al. [49] found no significant differences when comparing particle size and glycemic response from breads made from refined wheat flour, coarse whole grain flour, and fine whole grain flour. 

In our study on human subjects, all GI values calculated had large standard deviations, about 50% of the mean, which is common for these types of studies. Flavel et al. [50] reviewed the contributions of the testing methodology to the variability in the glycemic index of foods. They reported that the international standard allows for a range of choices for GI testing rather than one standardized protocol, resulting in significant variations in GI testing methodologies. Areas of variation include glucose specification and reference food used, as well as amount of food and drink given during testing, blood sampling site, and the tools used to measure blood glucose concentration.

As mentioned in the Introduction, several factors can influence the GI of a given food. As these muffins were all made from the same sorghum flour, the differences in the GI could not have resulted from differences in phenolic content, protein, etc. Food structure has been found to influence the GI of foods, and the differences found here among the sorghum muffins may indicate differences in the microstructure of these muffins. Further research investigating relationships between the physical properties of sorghum flour (e.g., particle size), flour composition, food structure, and GI are warranted. If differences in the GIs of the sorghum flour muffins are related to differences in food structure, this may be a unique way to control the GI in sorghum-based baked foods. This would be especially interesting when using sorghum lines known to have high levels of phenolic compounds.

Comparing all the flours ground using the 0.5 mm screen, the muffins made from brown rice had the lowest GI (Table 4), which was lower than that of the white-grained sorghum flour used in this study when ground through the same-size screen. Across all the sorghum flour sizes, the finely milled brown rice flour had a GI similar to that of the intermediate milled sorghum flour, and both of these flours had the lowest GIs of all samples. Each GI value is reported according to the blood glucose response to 20 g of available carbohydrate. A 20 g sample was used as opposed to the standard 50 g sample because this amount is more representative of what one may consume in a typical sitting. 

The greater the starch damage in flour, the more susceptible the damaged granules to enzymatic attack [51]. Starch damage is also associated with an increase in water absorption by starch granules [52]. Alpha-amylase, both from salivary glands and the pancreas, cleaves α-1,4 glucose bonds in amylose during digestion. Amylose becomes more readily digestible in a food as it is processed. Heat and hydration rupture starch granules and facilitate enzyme hydrolysis [53]. These factors suggest a possible connection between starch damage and glycemic response; however, no significant correlation between the amount of starch damage and glycemic response was present (Figure 2).

## 4. Conclusions

Muffins made from the whole-grain sorghum flour used in this study fell in the GI range of low to intermediate, depending on the particle size of flour used. Intermediately milled sorghum flour had in the lowest glycemic response when comparing the three particle sizes of sorghum flour used in this study. This may reflect differences in food structure when baked into a muffin. When compared to other grains tested, sorghum products from intermediately ground flour exhibited a significantly lower glycemic response, with the exception of brown rice flour. This points to the possible advantage of using sorghum flour in gluten-free foods for delivering products with a lower GI than corn or wheat. No correlation was found between the percentage of starch damage in flour and the GI of the baked product tested. These results can assist in the product development process to advance the quality of healthy, gluten-free, sorghum-based foods for consumers. Further research should investigate if these results can be duplicated and the possible reason for the lower GI of intermediate-particle-size sorghum flour.

## Figures and Tables

**Figure 1 foods-12-04188-f001:**
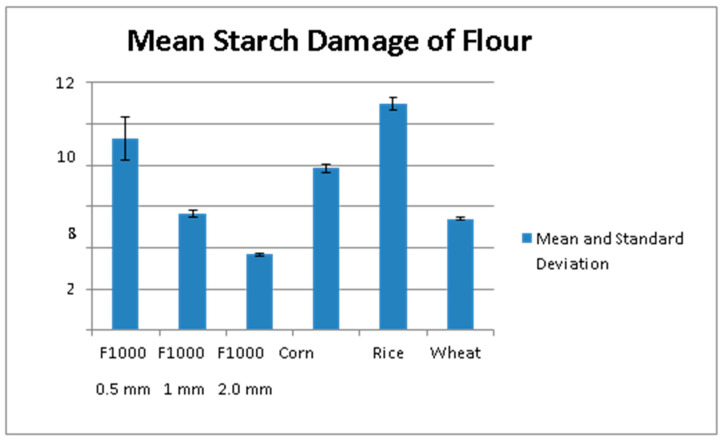
Percentage of starch damage among finely milled sorghum flour, intermediate milled sorghum flour, coarsely milled sorghum flour, corn flour, brown rice flour, and whole wheat flour.

**Figure 2 foods-12-04188-f002:**
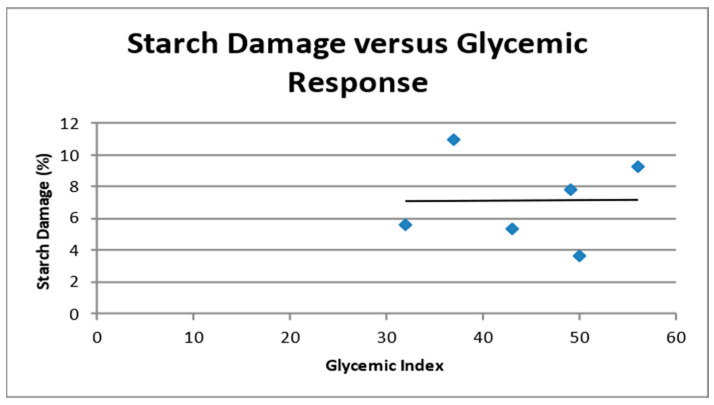
Correlation between percentage starch damage and glycemic index of finely milled sorghum flour, intermediate milled sorghum flour, coarsely milled sorghum flour, corn flour, brown rice flour, and whole wheat flour.

**Table 1 foods-12-04188-t001:** Muffin formula eater levels.

Flour Type	Water Amount (g)
Sorghum (Fine)	415 g
Sorghum (Intermediate)	350 g
Sorghum (Coarse)	315 g
Corn	400 g
Rice	385 g
Whole Wheat	385 g
All-Purpose	375 g

**Table 2 foods-12-04188-t002:** Adjusted serving sizes of muffins made from different flours to provide 20 g of available carbohydrates per muffin.

Flour Type	Serving Size (g)
Sorghum (Fine)	53.3
Sorghum (Intermediate)	47.8
Sorghum (Coarse)	55.0
Corn	49.1
Rice	44.3
Whole Wheat	55.0
All-Purpose	45.0

**Table 3 foods-12-04188-t003:** Mean d90 particle size distribution and mean particle size distribution of milled flours from various grains.

Flour Type	Mean d90 Flour Particle Size Distribution (μm)	Mean Particle Size Distribution (μm)
Sorghum (fine)	190.9 ± 0.6	82.2 ± 1.1
Sorghum (intermediate)	394.4 ± 11.4	166.9 ± 4.0
Sorghum (coarse)	731.7 ± 42.6	303.3 ± 15.5
Corn (fine	246.6 ± 47.1	92.6 ± 16.1
Rice (fine)	154.2 ± 2.6	67.9 ± 0.4
Wheat (fine)	307.4 ± 2.2	325.1 ± 4.5

**Table 4 foods-12-04188-t004:** Percent composition of 100 g muffin treatments made from milled flours of various grains.

Muffin Type	%Moisture	% Crude Protein	% Crude Fat	% Crude Fiber	% Ash	% Total Carbohydrate
Sorghum (fine)	55.62 ± 0.20 ^a^	4.38 ± 0.20 ^e^	0.79 ± 0.01 ^c^	0.38 ± 0.01 ^c^	1.33 ± 0.01 ^cd^	37.50 ± 0.20 ^e^
Sorghum (intermediate)	50.25 ± 0.04 ^c^	4.87 ± 0.05 ^d^	0.97 ± 0.02 ^b^	0.52 ± 0.03 ^b^	1.54 ± 0.01 ^ab^	41.80 ± 0.14 ^c^
Sorghum (coarse)	55.47 ± 0.01 ^a^	5.05 ± 0.02 ^c^	1.02 ± 0.01 ^b^	0.70 ± 0.02 ^a^	1.34 ± 0.02 ^cd^	36.41 ± 0.00 ^f^
Corn (fine)	52.73 ± 0.21 ^b^	3.74 ± 0.01 ^g^	1.17 ± 0.00 ^a^	0.28 ± 0.02 ^d^	1.38 ± 0.02 ^c^	40.72 ± 0.18 ^d^
Rice (fine)	48.81 ± 0.07 ^d^	4.15 ± 0.03 ^f^	0.29 ± 0.01 ^d^	0.06 ± 0.0 ^e^	1.50 ± 0.02 ^b^	45.20 ± 0.07 ^a^
Wheat (fine)	52.96 ± 0.05 ^b^	8.05 ± 0.05 ^a^	0.31 ± 0.00 ^d^	0.65 ± 0.02 ^a^	1.58 ± 0.01 ^a^	36.40 ± 0.03 ^f^
All-purpose	47.71 ± 0.10 ^d^	6.80 ± 0.02 ^b^	0.14 ± 0.00 ^e^	0.02 ± 0.00 ^e^	1.31 ± 0.01 ^d^	44.00 ± 0.07 ^b^

Numbers with different superscripts are significantly different (*p* < 0.05).

**Table 5 foods-12-04188-t005:** Mean glycemic response values to muffin treatments made from milled flours of various grains.

	Sorghum (Fine)	Sorghum (Intermediate)	Sorghum (Coarse)	Corn (Fine)	Rice (Fine)	Wheat (Fine)	All-Purpose Flour
GI	56 ± 33 ^b^	32 ± 17 ^a^	50 ± 26 ^ab^	49 ± 29 ^ab^	37 ± 18 ^a^	43 ± 23 ^ab^	44 ± 22 ^ab^

Numbers with different superscripts are significantly different (*p* < 0.05).

## Data Availability

Data supporting the reported results can be found in K-State Libraries, https://krex.k-state.edu/bitstream/handle/2097/13810/AshleyPruett2012.pdf?sequence=1&isAllowed=y (accessed on 8 November 2023).

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
