# Peer review of "Effect of Flour Particle Size on the Glycemic Index of Muffins Made from Whole Sorghum, Whole Corn, Brown Rice, Whole Wheat, or Refined Wheat Flours†"

_foods, 2023, doi:10.3390/foods12234188_

Round 1

Reviewer 1 Report

Comments and Suggestions for Authors

The article "Effect of Flour Particle Size on the Glycemic Index of Muffins Made From Whole Sorghum and Comparison to Muffins Made from Whole Corn, Brown Rice, Whole Wheat, or Refined Wheat Flours" submitted for review concerns the determine of the glycemic index of muffins made from whole-grain sorghum flour ground to three different particle sizes (fine, intermediate, course) and to compare the glycemic index to whole grain corn, wheat, and rice flours produced using the same product formula. The manuscript is well-written, clear, relevant to the field, and presented in a well-organized manner. The literature cited is mainly recent, and relevant publications include 3 references. The experimental design is adequate to test the hypothesis. However, the MS needs after addressing the following corrections to improve the quality of the article.

Abstract: it should give some details about the treatments, and they bought some numerical data, not only general trains, and provide a good conclusion and further outlook of the next status.

Introduction: the content of the introduction should be increased a little in the section on sorghum benefits because there are a lot of sorghum benefits, and many trials about the health benefits could be mentioned here to give the reader an overview of the main cohort of this study.

Materials and methods:

-       in the ingredient sections, you should provide the scientific names of all of the used plants.

-       Regarding the Starch damage protocol, please give the means of details of the determination of the starch damage using Megazyme kit.

-       In the Muffin Formulation and Preparation, please define salt as edible salt.

-       Please clarify how to determine the suitable water portion during the kneading of your formulas. Did you determine the absorption percentage of your flour?

Results and discussion:

-       Please follow the journal instructions regarding figures and tables

-       In Table 4, the first column disappeared; please check.

-       In all tables in the result and discussion section, please set all decimal digits in the same manner: one or two decimal digits.

-       In order to write a clear conclusion and to clarify and explain your work statements, It could be much better to do such a regression correlation between the glycemic index and the proximate chemical composition as well as with starch damage.

-       You said, “No correlation was found between percent starch damage in flour and GI of the baked product tested.” please write the correlation data (in a Table) and which kind of message you used to interpret your data.

Conclusion:

Please improve the conclusions. In its current form, the conclusions repeat the information in the abstract. In this part of the article, please include a brief verification of the hypothesis and suggest directions for further research.

Author Response

Reviewer 1 had good comments that have been all addressed as indicated below:

Abstract: it should give some details about the treatments, and they bought some numerical data, not only general trains, and provide a good conclusion and further outlook of the next status

Abstract has been re-written with more details about the treatments, more numerical data and a comment about conclusion and implications.

Introduction: the content of the introduction should be increased a little in the section on sorghum benefits because there are a lot of sorghum benefits, and many trials about the health benefits could be mentioned here to give the reader an overview of the main cohort of this study.

A few relevant articles added

Materials and methods:

-       in the ingredient sections, you should provide the scientific names of all of the used plants.

            Scientific names added

-       Regarding the Starch damage protocol, please give the means of details of the determination of the starch damage using Megazyme kit.

            Added

-       In the Muffin Formulation and Preparation, please define salt as edible salt.

            Done

-       Please clarify how to determine the suitable water portion during the kneading of your formulas. Did you determine the absorption percentage of your flour?

            Explained

Results and discussion:

-       Please follow the journal instructions regarding figures and tables

            Done

-       In Table 4, the first column disappeared; please check.

            Done

-       In all tables in the result and discussion section, please set all decimal digits in the same manner: one or two decimal digits.

Done, except for Table 4 since Glycemic Index values are usually reported as whole numbers

-       In order to write a clear conclusion and to clarify and explain your work statements, It could be much better to do such a regression correlation between the glycemic index and the proximate chemical composition as well as with starch damage.

This was not an objective of the study since there are a lot data listing the GI of various grains, which is assumed to be dependent on proximate composition, particularly total starch and fiber content. Also please note that all treatments were administered as 20 g available carbohydrates, thus making a correlation of GI to proximate analysis of the flours irrelevant

-       You said, “No correlation was found between percent starch damage in flour and GI of the baked product tested.” please write the correlation data (in a Table) and which kind of message you used to interpret your data.

The correlation coefficient was very low at only 0.15.  I have added a graph.

Conclusion:

Please improve the conclusions. In its current form, the conclusions repeat the information in the abstract. In this part of the article, please include a brief verification of the hypothesis and suggest directions for further research.

Done

Reviewer 2 Report

Comments and Suggestions for Authors

Research presented in the manuscript investigates the effect of flour particle size on glycemic index of „muffins“ made from different flour types.

Title of the work is too long and needs rewritting.

Abstract section needs more results to be presented.

Introduction section is appropirate with adequate referencing.

Material and method section has some issues that need addressing.

Results and discussion section needs a lot of corrections. Numerous missing data.

Conculsion section needs some additional informations derived from presented results.

References section needs major corrections.

Main issue with the scientific concept of this manuscript is conducted reseach on undefined bakery product type.

All detailed comments are noted in manuscript pdf file.

Comments on the Quality of English Language

Minor editing of English language required

Author Response

Reviewer 2 had good comments that have been all addressed as indicated below:

Title of the work is too long and needs rewriting.

Done

Abstract section needs more results to be presented.

Done

Introduction section is appropriate with adequate referencing.

Thanks

Material and method section has some issues that need addressing.

 Addressed as indicated

Results and discussion section needs a lot of corrections. Numerous missing data.

Corrected

Conclusion section needs some additional information derived from presented results.

Done

References section needs major corrections.

Corrected

Main issue with the scientific concept of this manuscript is conducted research on undefined bakery product type.

The hypothesis of this study had nothing to do with a bakery product as such. The “muffins” were made from a very simple formula only as carriers of the grain flours to be tested for their glycemic index

Line 91 Only 3 of the muffin treatments are shown.

Picture removed

Line 92 Why no mixing equipment was used

Because the amounts were so small that it would have created a bigger chance for errors than mixing by hand

Line 95 Change temperature units

Done

Table 2 Title

Adjusted serving size of muffins made from various flours to provide 20 g of available carbohydrates per muffin.

(Also serving size in the table corrected from 50 g to 20 g)

Table 3 Title

Mean d90 particle size distribution and mean particle size distribution of milled flours from various grains

Table 4 Title

Mean glycemic response values to muffin treatments made from milled flours of various grains 

Reviewer 3 Report

Comments and Suggestions for Authors

Thank you for submitting the manuscript "Effect of Flour Particle Size on the Glycemic Index of Muffins Made From Whole Sorghum and Comparison to Muffins Made from Whole Corn, Brown Rice, Whole Wheat, or Refined Wheat Flours." to Foods. The manuscript is interesting and is within the scope of Foods. However, it needs to be reorganized to improve clarity and there are typos and grammatical/technical issues that need to be corrected. Also, there are questions regarding methodology and data quality that need to be answered. Additionally, I have other suggestions:

- I suggest improving the title, making it more objective (and also smaller) and focused on the result obtained that is of greater importance.

- Line#14: preventing? I don't believe this is right.

- Line#16: course

- Line#17: I don't think the formulation can be the same since the flours are replaced.

- Abstract: overall, needs to be rewritten. Consider including more obtained results, especially numerical results.

- Keywords: consider changing the keywords to keywords that are not present in the title.

- Line#121: I believe that a subtitle that would better explain what was done is "In vivo protocol" and the first subitem would be "2.4.1 Subjects" and so on.

- Line#130-134: I suggest making a diagram of how the study was carried out. The way it is written, it seems that the same research subject received more than one type of formulation. That's it?

- Line#151: was the result of the human study assessed for normality? Why was this statistical test used? What statistical test was performed with the results of the proximal composition and other muffin analyses?

- Line#188: it is necessary to characterize the group of individuals that were used in the in vivo study. Do they have diabetes? What is the average glycemic index of this study outside of the study? And other intrinsic characteristics that make a difference when evaluating a group of living beings.

- The results as a whole need to be organized. Consider initially bringing the results of the muffin characterization and then the in vivo study.

- Line#214: what size of the surface area was studied for the present work? Where is this result?

- Figures: need to be improved in terms of technical aspects (not even the text font is standard with the article) and quality.

Comments on the Quality of English Language

Manuscript needs to be reorganized to improve clarity and there are typos and grammatical/technical issues that need to be corrected. 

Author Response

Reviewer 3 had good comments that have been all addressed as indicated below:

I suggest improving the title, making it more objective (and also smaller) and focused on the result obtained that is of greater importance.

Done

- Line#14: preventing? I don't believe this is right.

Corrected by removing the word “preventing”

- Line#16: course

Corrected to Coarse

- Line#17: I don't think the formulation can be the same since the flours are replaced.

Corrected to “similar”

- Abstract: overall, needs to be rewritten. Consider including more obtained results, especially numerical results.

Abstract re-written and numerical results added

- Keywords: consider changing the keywords to keywords that are not present in the title.

We disagree with this suggestion since keywords should inform a potential reader about the subject matter covered in a research paper

- Line#121: I believe that a subtitle that would better explain what was done is "In vivo protocol" and the first subitem would be "2.4.1 Subjects" and so on.

Corrected

- Line#130-134: I suggest making a diagram of how the study was carried out. The way it is written, it seems that the same research subject received more than one type of formulation. That's it?

We believe that no diagram is needed since the description given clearly indicates that subjects did receive more than one treatment: “Subjects received two treatments per week by random order with at least a 48h period in between. Each treatment was given twice as well as one dosage of a 20 g dextrose drink (reference food) over an eight-week period”

- Line#151: was the result of the human study assessed for normality? Why was this statistical test used?

What statistical test was performed with the results of the proximal composition and other muffin analyses?

“Two replications of each treatment were used in a randomized block design. GI values were analyzed using SAS, Software Release 9.3 (SAS, Institute Inc., Cary, NC, 2011). Difference in response between grains, flour particle size and proximate analysis composition of muffins was analyzed for significance. When treatment effects were found significantly different, the least square means with Dunnet’s P-values were used to differentiate treatment means. A level of significance was reported at p<0.05”.

- Line#188: it is necessary to characterize the group of individuals that were used in the in vivo study. Do they have diabetes?

The description clearly states that the subjects were healthy, i.e. no diabetes

“Eight healthy volunteers (ages 18-40) participated in the study”

What is the average glycemic index of this study outside of the study?

We don’t understand this comment! Glycemic Index refers to a food not a human subject.

And other intrinsic characteristics that make a difference when evaluating a group of living beings.

We were not evaluating the health of a group of living beings. We were measuring the glycemic index of different flours using “Eight healthy volunteers (ages 18-40)”, a standard procedure used in evaluating GI of foods.

- The results as a whole need to be organized. Consider initially bringing the results of the muffin characterization and then the in vivo study.

Done

- Line#214: what size of the surface area was studied for the present work? Where is this result?

“The incremental area under the glycemic response curve (iAUC) was constructed using the trapezoid model with fasting levels as the baseline by GraphPad Prism 5 software (GraphPad Software, Inc., La Jolla, CA). Area below the baseline was excluded”

- Figures: need to be improved in terms of technical aspects (not even the text font is standard with the article) and quality.

Done

Round 2

Reviewer 1 Report

Comments and Suggestions for Authors

The new version has been improved.

Reviewer 2 Report

Comments and Suggestions for Authors

The quality of the work is improved